# Efficient Representation Learning for Higher-Order Data with Simplicial Complexes

**Ruochen Yang**
University of Southern California
ruocheny@usc.edu

**Frederic Sala**
University of Wisconsin–Madison
fredsala@cs.wisc.edu

**Paul Bogdan**
University of Southern California
pbogdan@usc.edu

## Abstract

Graph-based machine learning is experiencing explosive growth, driven by impressive recent developments and wide applicability. Typical approaches for graph representation learning predominantly focus on pairwise interactions, while neglecting the patterns of higher-order interactions common to complex systems. This paper explores many-body interaction models, centering on simplicial complexes. From a theoretical point of view, we offer a pair of insights illustrating why higher-order models are necessary, why non-graph-based models generally cannot generalize well, while graph-based models may be able to do so. We conduct experiments on synthetic data, co-citation networks, co-authorship networks and gene-disease associations and show that simplicial complexes with certain relaxations can more efficiently capture underlying higher-order structures than non-graph structure, regular graph, hypergraph, and traditional simplicial complex-based learning frameworks.

## 1 Introduction

Graphs have emerged as popular and efficient tools to model complex structures and relationships in biological, chemical, social interactions, cyber-physical systems and many more types of systems. In a graph, nodes represent elementary units and edges encode the interactions of two entities [1–3]. While graph learning methods (e.g., graph neural networks) offer excellent performance in representation learning, predicting structure, and other tasks, these techniques often ignore higher-order relationships.

Regular graphs and pairwise interactions fail to capture group aspects where relationships and interactions are irregular and can appear among three or more components. For example, a graph cannot represent and distinguish the following two cases of co-authorship relations: (1) three authors collaborate together on one work, and (2) they pairwise co-author with each other. The two instances are both modelled as three fully-connected nodes in a regular graph regardless of the physical difference. Regular graph structures compress and collapse higher-order interactions to dyadic relationships and therefore lose high-dimensional information. To capture such complex relationship and avoid lossy representations, we must go beyond graphs and pairwise connections.

Simplicial complexes and hypergraphs—higher dimensional analogs of graphs—are two of the most intuitive and natural ways to represent group or collective interactions [3, 4]. Much of the earlier literature focuses on hypergraphs and develops representation learning frameworks that generalize graph neural networks (GNNs) [5–10]. In contrast, simplicial complexes build on the machinery of algebraic topology and enable us to define higher-order (collective) interaction analogs to the graph Laplacian [2, 3, 11]. They are also inherently imbued with hierarchical representations and rich algebraic structure which may be missed by hypergraph descriptions [12, 13]. For this reason,

R. Yang et al., Efficient Representation Learning for Higher-Order Data with Simplicial Complexes. *Proceedings of the First Learning on Graphs Conference (LoG 2022)*, PMLR 198, Virtual Event, December 9–12, 2022.

simplicial complex-based models have recently been proposed [14–18]. Despite that these approaches are well defined for simplicial complexes, the majority of the experiments still utilize higher-order data lifted from pairwise interactions, meshes, images and trajectories, while neglecting naturally-built many-body interactions which are heterogeneous and irregular in the structure.

In the rest of the paper, we first discuss the recent works on how to generalize graph representation learning with simplicial complexes in section 2. In section 3, we propose to use a relaxation of the formal simplicial complexes to capture irregular higher-order data. We describe an efficient and flexible learning framework which is more suitable for diverse and information-rich structures based on higher-order Laplacians and adjacency matrices. We also provide theoretical insights in section 4, based on a simple but general data generation process, demonstrating the necessity of accounting for higher-order interactions—and how this interacts with generalization. In section 5, we consider a variety of types of many-body interaction data including synthetic data, clique complexes built from regular graph, and naturally built higher-order data in the form of simplicial complexes with certain relaxation. The results show that relaxed simplicial complex based learning models can efficiently capture the higher dimensional information and surpass existing graph learning methods on simplex classification tasks, outperforming the best baseline by up to $6.7\%$ in accuracy.

## 2   Related Works

Unsupervised representation learning methods [14, 16, 17] extend *node2vec* embeddings [19] to simplicial complexes with random walks on interactions through Hasse diagrams and simplex connections inside $p$-chains. In the recent three years, studies focus more on the semi-supervised learning on simplicial complexes which generalizes graph neural networks. Simplicial neural networks (SNN) [12] generalize spectral graph convolution [20] to simplicial complexes with higher-order Laplacian matrices. Yang et al. [21] further propose the simplicial convolutional neural networks (SCNN) with simplicial filters to exploit the lower- and upper-neighborhood relationships. In Bunch et al. [15], the authors propose a simplicial 2-complex convolution layer, but with limits on the maximum dimension of higher-order data and on its application to images. Hajij et al. [16, 22, 23] propose encoder-decoder and message passing based representation learning models such as convolutional cell complex networks (CCXN) on simplicial complexes and cell complexes. Bodnar et al. [18, 24] propose message-passing simplicial networks (MPSN), simplicial isomorphism networks (SIN) and cell isomorphism networks (CIN), which can distinguish strongly regular graphs, classify trajectories and graphs. Roddenberry et al. [25], discuss the permutation, orientation equivariance and simplicial awareness properties of simplicial neural architectures and propose SCoNe for trajectory prediction. Within the last year, the attention mechanism is employed to generate representations on simplicial complexes and combinatorial complex [13, 26–28] and the Hodge Laplacian is exploited to learn knowledge of graph structures [29–31].

Although the aforementioned models are well-defined on general simplicial complex structures, most of the models are examined only on analogs of real-world complex higher-order information, which is built from images [15, 27], meshes [22, 28], trajectories [18, 25–27], pairwise interactions (graphs) [13, 14, 18, 23, 24, 28] or synthetic random models [14, 17, 23, 25–27]. One naturally-built higher-order dataset (co-authorship) is examined [12, 21, 26], but the data source of the simplicial complex is restricted as it is constructed as subsets of only 80 papers. A comprehensive analysis on complex naturally-built many-body interactions is still lacking.

Our motivation is to understand what makes higher-order/simplicial complex-based frameworks perform well in practice and in theory. We wish to go beyond traditional graph representation learning and focus on higher-order data-based tasks such as simplex classification with practical and efficient data structure. Specifically, we want to tackle a wide variety of simplicial complexes including rich and organic higher-order data to capture the heterogeneous many-body interactions which commonly exist in real world.

## 3   Backgrounds

### 3.1   Higher-Order Data as Simplicial Complexes

**Definitions.** A simplicial complex generalizes a graph by accounting for higher-dimensional information. The interaction among points (nodes) is characterized by a simplex [11, 32]. An oriented

$p$-dimensional simplex $\sigma$ is composed of $(p+1)$ points and is denoted by $\sigma = [i_0 \dots i_p]$; it represents an interaction among a group of points. For example, a 0-simplex is a node, a 1-simplex is an edge, a 2-simplex is a triangle, a 3-simplex is a tetrahedron, and so on. Going beyond pairwise interactions, a simplex can differentiate among interactions with different dimensions.

A simplicial complex refers to a set of simplices. A $p$-chain is the finite formal sum of $p$-simplices and the group of $p$-chains on simplicial complex $X$ is denoted by $C_p(X)$. If the points in a $p$-simplex $\sigma$ are the subset of the points in a $(p+1)$-simplex $\tau$, where only one element is omitted, then $\sigma$ is called a face of $\tau$ and $\tau$ is called a coface of $\sigma$. Boundary map $\partial_p : C_{p+1}(X) \to C_p(X)$ indicates the existence / orientation of each $p$-simplex as a face of each $(p+1)$-simplex. The boundary map $\partial_p$ is described by incidence matrices $B_p$ of dimension $N_p \times N_{p+1}$, where $N_p$ is the number of $p$-simplices in the simplicial complex. The higher-order Laplacian $L_p$ describes the diffusion on a $p$-chain and generalizes graph Laplacians [11]. The $p$-order Laplacian matrix is calculated from incidence matrices by the formula $L_p = B_{p-1}^T B_{p-1} + B_p B_p^T$ when $p > 0$. The 0-order Laplacian is $L_0 = B_0 B_0^T$.

**Handling Higher-Order Data with Relaxed Simplicial Complexes.** The conventional definition of simplicial complexes requires them to be closed under taking subsets. This presents a challenge for models that operate on higher-order data via simplicial complexes: (1) Taking subsets will cause blowup when high-dimension simplices are present in large-scale datasets. This is a common phenomenon especially in real-world systems where interactions and group behaviors are sophisticated and irregular. For example, the co-authorship complexes built from Semantic Scholar in Ebli et al. [12] have 25,000 and 100,000 simplices, but they are constructed as subsets of only 80 papers where some of them are co-authored by 10 researchers. This inclusion restriction will cause the datasets to exponentially increase and crucial information in naturally-existed simplices will be obscured by the potential redundancy embedded in the subsets. (2) It is difficult to explain the physical meaning and properties of simplices which are added as subsets of higher-order instances. For instance, a co-authorship map can be considered as a simplicial complex where points are authors and simplices are papers with venue as label and word embedding as feature. If two scientists $a$ and $b$ work together with different third researchers multiple times (2-simplex $[a, b, c]$ and $[a, b, d]$) but never exclusively coauthor with each other, how can we understand the existence of the 1-simplex $[a, b]$ added due to the inclusion?

We wish to practically and efficiently capture many-body interactions. In [28], the authors introduce combinatorial complexes which allows arbitrary set relations to generalize simplicial complexes. Although the combinatorial complex does not require downward closure as in simplicial complexes, the method is examined on datasets where the inclusion property still preserves. In this work, we consider a relaxation of the conventional definition and allow simplicial complexes to potentially not be closed under subsets. We further discuss the relationship of relaxed simplicial complexes and hypergraphs and explain our choice of relaxed simplicial complexes in Appendix C. In the rest of this paper, we use $p$-chain to refer the set of $p$-simplices, where $p$-simplices are given as groups of interactions in the dataset. In addition, we want to use simplex to represent an activity among several components (e.g., a paper written by several co-authors), so we take simplices to be unoriented and all elements in the incidence matrices to be non-negative. These simple adjustments will enable representation learning frameworks to be more flexible on a wide variety of higher-order datasets and able to accommodate large-scale data while avoiding the size explosion problem.

## 3.2 Representation Learning Models with Relaxed Simplicial Complexes

Among our goals are to analyze topological structure and to examine message passing and aggregation methods on relaxed simplicial complexes. We first describe a notion of *connection* for simplices and then higher-order adjacency matrices. This allows the heterogeneous structure of simplicial complexes to be associated with various powerful graph machine learning models. Afterwards, we describe *simplex convolutional networks* (SCN) and *simplicial complex convolutional networks* (SCCN) models. These models exploit the generalization of the graph convolution operation [33]. Using the same principles, we also consider *sc2vec*, a latent representation learning framework for simplicial complexes.

**Connection of Simplices and Higher-Order Adjacency Matrices.** To define the connection of two $p$-simplices, we utilize the higher-order adjacency matrix $A_p$ of $p$-chains with the help of the higher-order Laplacian and incidence matrices. Recall that the $p$-order Laplacian is the sum of two

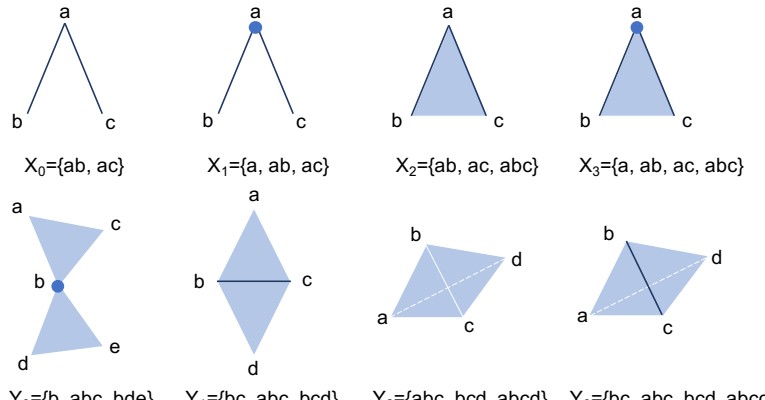

**Figure 1:** Top: Different connections of 1-simplices (edges) $ab$ and $ac$. In $X_0$, $A_1(ab, ac) = 0$ because $ab$ and $ac$ neither share a face nor a coface. In $X_1$, $A_1(ab, ac) = 1$ since $ab$ and $ac$ share a face $a$. In $X_2$, two 1-simplices share a coface $abc$ and, thus, $A_1(ab, ac) = 1$. $ab$ and $ac$ have the same face $a$ and coface $abc$ in $X_3$, so $A_1(ab, ac) = 2$. Bottom: Different relationships of 2-simplices (triangles) $abc$ and $bcd$ (or $bde$). Similarly, in $Y_0$, the connection of $abc$ and $bde$ is 0. In $Y_1, Y_2, Y_3$, the element in the 2-order adjacency $A_2(abc, bcd)$ is 1, 1, 2, respectively.

parts, the lower Laplacian $L_p^{lower} = B_{p-1}^T B_{p-1}$, and the upper Laplacian $L_p^{upper} = B_p B_p^T$. The lower and upper $p$-th Laplacian, respectively, describe the relationship of $p$-simplices through faces and cofaces [15, 16, 18]. We use the $(i, j)$-th element in the $p$-order adjacency matrix to denote the connection of two $p$-simplices $\sigma_{p,i}$ and $\sigma_{p,j}$ $(i \neq j)$:

$$A_p(i, j) = L_p(i, j) = \mathbb{I}\{\sigma_{p,i}, \sigma_{p,j} \text{ share a face}\} + \mathbb{I}\{\sigma_{p,i}, \sigma_{p,j} \text{ share a coface}\}. \quad (1)$$

When $i = j$, we assign $A_p(i, j) = 0$. Figure 1 shows various connecting relationships for two 1-simplices (top) and two 2-simplices (bottom). Note that subsets of simplices are not necessarily in the simplicial complex without the inclusion rule. For example, if $abc$ is in the simplicial complex, $ab$, $ac$ and $bc$ are not automatically included, unless otherwise stated.

**Neural Network-based Representation Learning for Simplicial Complexes.** Following the graph convolutional network (GCN) formalism [33], SCCONV and CCXN which generalize GCN to simplicial complex and cell complex [15, 16], we first describe *simplex convolutional networks* (SCN). We introduce the following notations: $\psi$ is a non-linear activation function, $H_p^{(l)} \in \mathbb{R}^{N_p \times E_p^{(l)}}$ is the simplex embedding of $p$-chains as the input of layer $l$, $N_p$ is the number of all $p$-simplices, and $E_p^{(l)}$ is the embedding dimension. $H_p^{(0)} = F_p$ represents the features of the $p$-chain. $W_p^{(l)}$ is the trainable weight. Self-loops with strength 2 are added to the $p$-order adjacency matrix $A_p$: $\tilde{A}_p = A_p + 2I_{N_p}$ for $p = 1, 2 \ldots p_{max} - 1$. We choose the self-loop strength to be 2 because a simplex shares faces and cofaces with itself. When $p = 0, p_{max}$, it becomes $\tilde{A}_p = A_p + I_{N_p}$. The adjacency matrix is normalized by $\tilde{D}_p^{-\frac{1}{2}} \tilde{A}_p \tilde{D}_p^{-\frac{1}{2}}$, where $\tilde{D}_{p,ii} = \sum_j \tilde{A}_p(i, j)$. The convolutional layers are defined on each $p$-chain in the simplicial complex as follows:

$$H_p^{(l+1)} = \psi\left(\tilde{D}_p^{-\frac{1}{2}} \tilde{A}_p \tilde{D}_p^{-\frac{1}{2}} H_p^{(l)} W_p^{(l)}\right). \quad (2)$$

Compared with SNN [12], the SCN model is more scalable and can easily accommodate high-dimensional features, as the convolutional propagation rule is a localized first-order approximation of the spectral graph convolutional operation [33].

In SCN, an independent convolutional operation is applied to each $p$-chain. We also consider *simplicial complex convolutional networks* (SCCN), where connections of all simplices are examined regardless of the dimension as in [28]. We define the full adjacency matrix $A$ for the simplicial

complex as:

$$A = \begin{bmatrix} \alpha A_0 & \beta B_0 & 0 & \dots \\ \beta B_0^T & \alpha A_1 & \beta B_1 & \dots \\ 0 & \beta B_1^T & \alpha A_2 & \dots \\ \vdots & \vdots & \vdots & \ddots \end{bmatrix}. \tag{3}$$

The full adjacency matrix $A$ has each $p$-adjacency matrix $A_p$ on its main diagonal and incidence matrix $B_p$ on the first diagonal below and above. Here, $\alpha$ and $\beta$ are weights for different types of connections. The $p$-adjacency matrix $A_p$ captures the relationship of simplex within the same dimension (i.e, upper or lower connected), while the incidence matrix $B_p$ contains the connections of a $p$-simplex and a $(p-1)$-simplex when the $(p-1)$-simplex is a face of the $p$-simplex. SCCN exploits the convolutional operation in the same way as in equation (2) to the full adjacency matrix $A$. In SCCN, simplices whose dimension has limited samples can be better learned, which is especially beneficial for high-dimensional cases where samples are usually less than low dimension.

Note that in this work, we do not include the possible connection of two simplices when the difference of their dimensions is larger than one. However, the definition of the full adjacency matrix in equation (3) can be easily modified to have the connection of two simplices of arbitrary dimension difference $d$ by assigning $B_p B_{p+1} \dots B_{p+d}$ on the corresponding $(d+1)$ diagonals. We further discuss the choice of equation (3) in Appendix D.

**Latent Representation Learning for Simplicial Complexes.** From a practical perspective, there are many cases where we only have access to pure interaction information. In other words, our data consists of structure without any features. We seek a model that handles such data as well. Simplex2vec [14], cell2vec [16], k-simplex2vec [17] and SCA [22] are proposed to learn latent representation from simplicial complexes and cell complexes. Following a similar idea of applying a node2vec-style approach to the full adjacency matrix in equation (3), we consider a latent representation learning model *sc2vec*. Details of sc2vec are presented in Appendix E.

## 4 Theoretical Insights

We provide a pair of theoretical insights related to simplicial complex-based models and higher-order graph models in general. Each relies on a proxy data generation model for graph-structured data. While simple, this model motivates the need for using graph-based models in multiple contexts. Our first insight is that higher-order distributions (representing the dependencies found in simplicial complexes) cannot be approximated by lower-order ones, motivating the use of higher-order models such as simplicial networks. Our second insight studies node / simplex classification with graph-structured data. We show that while it is possible to train a conventional (not graph-based) model that generalizes despite the numerous dependencies induced by graph-structured data, to ensure generalization it is necessary to certify that the dependencies are very weak. However, graph-based models directly rely on these dependencies, implying that generalization is possible, as we observe in practice.

### 4.1 Graph-Structured Data Model

We use the following as a proxy model for graph-based learning tasks. Set $X = (X_1, \dots, X_n)$, where $X_i \in \mathbb{R}^d$ are features and $Y = (Y_1, \dots, Y_n)$ with $Y_i \in \{\pm 1\}$ are labels. Let $G$ be a hypergraph with vertex set $V(G) = \{1, \dots, n\}$ and edge set $E(G)$. Then,

$$f_X(Y) = \frac{1}{Z} \exp\left( \sum_{i=1}^{n} X_i^T \theta Y_i + \beta \sum_{e \in E(G)} \prod_{v \in e} A_e Y_v \right). \tag{4}$$

where $\theta_i \in \mathbb{R}^d$, $\beta$, and the $A_e$ are model parameters, and $Z$ is a normalizing partition function .

In (4), the left-hand side term by itself is a linear model; it can be easily replaced with any other data model. The right-hand side term, however, introduces higher-order graph structure over the data; it promotes symmetries among labels. The $\beta$ parameter controls the importance of features versus dependencies. If we take $G$ to be a graph, so that the edges $e$ involve only two vertices, we obtain a model in $Y$ identical to the Ising model of Daskalakis et al. [34], which studied linear and logistic

regression with dependent data. The more general version (4) allows for more complex dependencies, including simplicial complexes and hypergraphs.

We are especially interested in the setting where we take the hyperedges $e$ to simulate a relaxed simplicial complex, as described in Section 3.1. For example, we can take $E$ in (4) to be $\{\{a\}, \{b\}, \{c\}, \{d\}, \{e\}, \{a, b\}, \{b, c\}, \{a, c\}, \{c, e\}, \{a, b, c\}, \{b, c, d\}\}$, yielding the complex in Fig. 2 in Appendix B. Note that this model captures node classification tasks, as the labels are attached to nodes. However, we could also construct a model, along the same lines for classifying simplices (as we do in our experiments).

## 4.2 Why does higher-order structure matter?

An initial question when studying higher-order models, like simplicial networks, is why one should bother with such models. After all, if a lower-order distribution can well-approximate a higher-order one, regardless of the structure or modeling choices, then certainly a lower-order model itself should suffice. We show this is not the case.

Concretely, higher-order structure can be arbitrarily important. Our result uses a small simplicial complex to show that no graph-based distibution can approximate it:

**Proposition:** Consider the class of models $\mathcal{F}$ of (4) with one or more higher-order interactions, including the class of simplicial complexes, and models $\mathcal{F}_\ell$ without such interactions (graph-based models). There exists $f \in \mathcal{F}$ so that for any $f' \in \mathcal{F}_\ell$, the divergence between $f$ and $f'$ is bounded away from zero. Specifically, for any $\delta > 0$, $d_{\mathrm{TV}}(f, f') \geq \frac{1}{4} - \delta$. $\qquad\square$

This result shows that there are distributions that cannot be approximated by lower-order ones; there can be a large constant gap in total variation distance between them. This suggests that we should use higher-order models, motivating our study of simplicial complex networks.

## 4.3 Why are non-GNNs insufficient?

Next, we explore the generalization ability of conventional models that do not incorporate graph structure when operating on points that are sampled from (4). We do so for the conventional node classification task. The main difference between a conventional setting for generalization and the graph-based data one is that the dataset is no longer i.i.d.; indeed, the labels may be highly dependent via the right-hand side of (4). Despite this challenge, it is still possible to show generalization by using techniques based on concentration in dependent settings, an exciting area with significant progress in the last decade [35–37]. Our goal is to study generalization result for node classification with graph-structured data. Our dataset is $S = \{(x_1, y_1), \ldots, (x_n, y_n)\}$, where $x_i \in \mathbb{R}^d$ and $y_i \in \{-1, +1\}$. These points are not i.i.d.; they are drawn from the distribution (4). We learn a function $f : \mathbb{R}^d \to \{-1, +1\}$. For a loss function $\ell$, e.g., the 0/1 loss, the risk is $R(f) = \mathbb{E}[\ell(f(x), y)]$ and its empirical counterpart is $\hat{R} = \frac{1}{n} \sum_{i=1}^{n} \ell(f(x_i), y_i)$. A standard result in the i.i.d. setting is the following Rademacher complexity bound [38]. With probability at least $1 - \delta$,

$$R(f) \leq \hat{R}(f) + \hat{\mathfrak{R}}_S(F) + 3\sqrt{\frac{\log 2/\delta}{2n}}, \tag{5}$$

where $\hat{\mathfrak{R}}_S(F)$ is the empirical Rademacher complexity for our model function class $F$ and dataset $S$. The i.i.d. requirement is needed for the use of McDiarmid's concentration inequality. Below, we relax this requirement.

**Dealing with dependencies.** The main technical challenge is that our dataset here is not i.i.d., since the labels $y$ are also connected via the graph / hypergraph / simplicial complex structure. If the dataset at minimum contains some degree of independence, it is possible to apply [39], which derives a variant of McDiarmid's inequality for a particular graph dependency structure. This dependency structure specifies which nodes are dependent (i.e., those connected by an edge) and which nodes are independent (those which are not neighbors). However, this assumption is potentially too strong for us: because of the longer-range dependencies in (4), we may not have *any* pairs of nodes which are independent.

On the other hand, many of the dependencies might be weak. This is likely to be the case for many applications of practical interest, where the features provide the majority of the signal and the graph-based dependencies provide the remaining portion. A powerful formalization of the concept of weak dependence is Dobrushin's condition [37], stated in terms of influences $I_{j \to i}(y)$. For $y = (y_1, \ldots, y_n)$, set the influence of $y_j$ on $y_i$ to be

$$I_{j \to i}(y) = \max_{y_{-i-j}, y_j, y_j'} d_{\mathrm{TV}}(P_{y_i | y_{-i}}(\cdot | y_{-i-j}, y_j),$$

$$P_{y_i | y_{-i}}(\cdot | y_{-i-j}, y_j')).$$

Here, $y_{-i-j}$ consists of the vector with all the entries of $y$ except indices $i$ and $j$. The basic intuition is to measure the maximum change in the distribution over $y_i$ when changing $y_j$ over all possible configurations of conditional distributions. If $\alpha(y) := \max_{1 \le i \le n} \sum_{j \neq i} I_{j \to i}(y) < 1$, then Dobrushin's condition is satisfied. Moreover, this permits the construction of a dependent version of McDiarmid's inequality [37]. Specifically, consider a distribution $P$ over $\{-1, +1\}^n$ satisfying Dobrushin's condition with coefficient $\alpha$ and a function $f : \{-1, +1\}^n \to \mathbb{R}$ with the bounded differences property $|f(y) - f(y')| \le \sum_{i=1}^n \mathbb{1}\{y_i \neq y_i'\}\lambda_i$ for a set of parameters $\lambda_1, \ldots, \lambda_n \ge 0$. Then, for all $t > 0$,

$$P(|f(y) - \mathbb{E}[f(y)]| \ge t) \le 2 \exp\left(-\frac{(1-\alpha)t^2}{2\sum_{i=1}^n \lambda_i^2}\right). \tag{6}$$

Using (6) to replace the standard i.i.d. version of McDiarmid's inequality will permit us to derive the Rademacher complexity bounds as in Mohri et al. [38]. Specifically, instead of (5), we now get

$$R(f) \le \hat{R}(f) + \hat{\mathfrak{R}}_S(F) + 3\sqrt{\frac{\log 2/\delta}{(1-\alpha)2n}}, \tag{7}$$

This is implicitly based on the influence matrix $\{I_{j \to i}(y)\}_{i,j}$. Worse, if $\alpha \ge 1$, then the correlations can be arbitrarily strong, and no concentration may result. To ensure such a non-GNN model generalizes, we must certify that $\alpha < 1$. However, as we observe, this is an extremely strict limitation. As a result, conventional models will often not suffice.

**Evaluating Dobrushin's condition in hypergraphs.** The influence matrix can be bounded by the dependency parameters in the model (4) in the following way. Suppose we are examining node $y_i$ and let all of the hyperedges that include it as a term be $e_1, e_2, \ldots, e_m$. Then, we have the following:

$$P_{y_i | y_{-i}}(\cdot | y_{-i-j}, y_j) = \frac{\exp(g_\beta(A, Y))}{\exp(g_\beta(A, Y)) + \exp(-g_\beta(A, Y))},$$

where $g_\beta(A, Y) = \beta \sum_k \prod_{v \in e_k} A_{e_k} Y_v$. To compute the $I_{j \to i}(y) = \max_{y_{-i-j}, y_j, y_j'}$, we can now use the formula above, yielding an expression for the influence in terms of $\beta$ and the $A_e$ adjacency matrix terms. While in general this does not yield clean bounds that can be used to easily state Dobrushin's condition, in special cases, it is possible to do so. For example, suppose that $G$ is a graph and $A_e = 1$ for all edges. Then, it was shown in Hayes [40] that $I_{j \to i}(y) \le \tanh(\beta)A_{ij}$, where $A_{ij} = 1$ for edges and 0 otherwise. We can show a generalization of this result for simplicial complexes, in the special case where the face weights have some regularity. Let $G$ be a simplicial complex on $p$ nodes with a single facet, with $A_e = 1$ for all $e$ excluding the facet, where we set $A_{\text{facet}} = 0$. Then, the influence term $\{I_{j \to i}(y)\}_{i,j}$ satisfies $I_{j \to i}(y) \le \tanh(\beta)$.

This implies that we can achieve Dobrushin's condition, and thus achieve generalization, if we ensure that $\beta < \tanh^{-1}(\alpha/(n-1))$, for any $\alpha < 1$—which is a very strong requirement.

However, models that do not rely on the i.i.d. assumption, such as graph-based models, including GNNs and higher-order variants, do not fall prey to such strict requirements for generalization.

Altogether, the two theoretical insights suggest that to handle non-i.i.d. data, we must use graph models of some order. In addition, among such models, to deal with higher-order dependencies, we must use higher-order models, such as the models of relaxed simplicial complex. This provides the theoretical motivation for our work; it is also consistent with empirical evidence we have observed.

In this section, we aim at providing an initial step towards understanding which models will generalize on data structured according to higher-order graphs. There are two steps here: (1) understanding

why simply modeling data according to standard graphs is insufficient and (2) understanding why particular networks generalize. Statistical learning theory has only taken very preliminary steps in this direction. Note that even a simple notion of generalization has not yet been agreed upon—unlike conventional cases, our data is not i.i.d., so that we cannot sample "new" points to test our trained model. This issue affects both regular graph-structured and higher-order graph-structured data. We bypass this issue by providing two types of results: first, a simple result showing the distinction between distributions on binary graph-structured data vs higher-order graph-structured data—which is applicable to any kind of model, and, second, a result applying the famous Dobrushin's condition, which enables generalization for at least some non i.i.d. cases. Indeed, the first result suggests that there are genuinely cases where no GNN will be able to ultimately perform well—but a higher order model, such as our proposed model, will. The second result suggests that there is at least a possibility of achieving some notion of generalization.

## 5    Experiments

We evaluate the representation learning framework with relaxed simplicial complexes on a wide variety of synthetic and real-word datasets for the simplex classification task. The prediction results show that the representation learning models with relaxed simplicial complexes formalism efficiently capture higher-order information in multiple datasets and outperform the best baselines by up to $6.7\%$ in accuracy.

### 5.1    Datasets

We apply the models on several synthetic complexes (Syn and SBM), clique complexes (Cora and Pubmed), and naturally-built simplicial complexes (DBLP, DisGe, PPI-BP and HPO-METAB). The statistics of the datasets can be found in Table 3 of Appendix F.

**Cora and Pubmed:** We take the benchmark citation datasets and build clique complexes. **Syn:** We take the structure of the Cora clique complex, randomly assign weights between $0.1$ and $1.0$ for each simplex, and then generate features and binary labels of points according to equation (4). We take simplices where inside points have the same label and use the point average as simplex features. **SBM:** We first generate a graph using the stochastic block model [41] with three categories of nodes (200 each), 0.08 intra-linking probability, 0.03 inter-linking probability, and then build clique complexes. **DBLP:** The DBLP co-authorship simplicial complex is constructed from the DBLP co-authorship hypergraph in Yadati et al. [5]. Points are authors and simplices are papers with labels representing the category of the venue and features are the word dictionary. **DisGene**[1]**:** We construct a simplicial complex where a simplex is a disease and points in a simplex are genes associated with the corresponding disease. The label of the simplex is the MeSH disease class. Features are built from the gene-disease relationship (disease type, pleiotropy index etc.). **PPI-BP:** In the molecular biology simplicial complex, each simplex is a collection of proteins in the same biological process and its label is the type of the biological process. **HPO-METAB:** The simplicial complex is built from a rare disease diagnosis dataset, where a simplex is a rare disease and the points are associated phenotypes. The label is the type of the disease. The structures of PPI-BP and HPO-METAB are built based on the subgraph in Alsentzer et al. [42].

Building simplicial complexes from regular graphs (Cora, Pubmed) can be viewed as a decompression process. A $(p+1)$-clique is considered to be a $p$-simplex if it is a maximal clique and all points in the clique have the same label. The simplex is labeled with the corresponding category and the feature is the average of node features inside the clique. By definition, subsets of a maximal clique will not be taken into consideration, and thus, may cause information loss in the decompression process. To capture the hierarchical structure of higher-order data and avoid the size explosion problem due to the inclusion rule and point combination in high dimension simplex, we only add the first-order sub-simplices (with $p-1$ points) of each $p$-simplex. Every node that has appeared in the maximal cliques or its first-order subsets are considered as 0-simplices. Nodes in the original graphs are discarded if they don't belong to any simplex. For DBLP coauthorship and DisGene, we take the set of many-body interactions existed in the original data as the simplicial complex without inclusion and orientation. For the datasets with pure topological informations (SBM, PPI-BP and HPO-METAB) which are used for latent representation learning models, we take all subsets while computing the

---

[1]http://www.disgenet.org/

higher-order adjacency matrix in order to compare with the baseline models (k-simplex2vec and simplex2vec), which both preserve the inclusion properties.

## 5.2 Baselines

In this work, we are interested in predicting the label of activities which have contributions from multiple components (simplices). We use the following representation learning models of hypergraphs, simplicial complexes, regular graphs, and non-graph structures as baseline models.

**DHE:** Payne [9] proposes to use the vertex and hyperedge embeddings as well as hyperedge features to perform hyperedge classification. **SIN:** Bodnar et al. [18] propose SIN for graph classification problem. Here, we replace the readout layer with an output layer to predict the label of each simplex. **Simplex2vec:** [14] This unsupervised representation learning approach adopts symbolic embeddings to compute the community structure on simplicial complexes via the Hasse diagram. **K-simplex2vec:** [17] This unsupervised representation learning framework extends the node embedding methods with biased random walks to simplices and considers the interaction of simplices in every $p$-chain. **GCN and MLP:** To show the importance of information embedded in higher-order structures, we implement graph convolutional networks (GCN) [33] and multilayer perceptron (MLP) models on the corresponding collapsed graph and non-graph data.

**Table 1:** Simplex classification accuracy (%) of the SCN, SCCN and baselines. Best accuracy is marked as bold.

| Dataset | Syn | Cora | Pubmed | DBLP | DisGene |
|---------|-----|------|--------|------|---------|
| SCN | 68.90 | 94.68 | 95.23 | 66.81 | **39.17** |
| SCCN | **74.95** | **95.65** | **96.84** | **75.69** | 34.90 |
| DHE | 68.22 | 86.46 | 94.50 | 67.38 | 36.79 |
| SIN | 60.26 | 80.89 | 93.04 | 69.60 | 36.66 |
| GCN | 62.83 | 92.61 | 95.02 | n/a | n/a |
| MLP | 58.58 | 89.47 | 90.30 | 73.95 | 36.63 |

## 5.3 Experimental Settings and Results

We apply the SCN and SCCN models with one hidden convolutional layer on the datasets described above. We chose the hidden dimension to be 16, ReLu as the activation function, and used the Adam optimizer [43] with learning rate 0.001 to train the SCN and SCCN models. A detailed description of the experimental settings is provided in Appendix section G.

We repeat each experiment of the SCN and SCCN model 100 times with shuffled train/validation/test splits and show the mean accuracy in Table 1. The SCCN model outperforms other baselines on the synthetic network, Cora and Pubmed co-citation simplicial complexes and DBLP co-authorship data. The SCN model also beats other baselines on the DisGene dataset.

**Table 2:** Simplex classification accuracy (%) of the latent representation embedding models. Best accuracy is marked as bold.

| Dataset | SBM | Cora | PPI-BP | HPO-METAB |
|---------|-----|------|--------|-----------|
| simplex2vec | 94.48 | 73.37 | 36.30 | 51.22 |
| k-simplex2vec | 97.43 | 90.55 | 30.33 | 25.10 |
| sc2vec | **100.00** | **93.86** | **36.90** | **56.18** |

In addition, we test the latent representation learning method sc2vec as well as the baselines on SBM, Cora, PPI-BP and HPO-METAB datasets. Here, features are not used. To examine the embedding performance, we use the simplex embedding results as the input to one-vs-rest logistic regression classifiers to predict the label of the simplex, following [19] and [44]. We repeat each experiment 50 times and present the mean accuracy of the multi-label classification in Table 2. The result shows that the embedding method sc2vec defined on the higher-order adjacent matrix also achieves excellent performance compared to the baselines.

We conclude that SCN, SCCN and sc2vec offer strong performance when performing representation learning of higher-order data via relaxed simplicial complexes. Complexity analysis, more experimental details and additional results of citation prediction are provided in the Appendix.

## 6 Conclusion

In this paper, we examined the representation learning framework with relaxed simplicial complexes meant to characterize the higher-order interactions embedded in real-world complex systems such as social and biological networks and cyber-physical systems. Theoretically, we showed that higher-dimensional dependencies cannot be modelled by regular graph-based networks and that conventional models cannot handle such dependencies either, in terms of generalization. The outstanding performance of the SCN, SCCN and sc2vec models on synthetic, clique complexes from graph, and naturally built simplicial complexes shows the efficiency in capturing high dimensional data using hierarchical models. Future work includes studying the relationship of simplices of arbitrary dimensions and embedding algebraic topology properties such as Betti number into the representation learning framework.

## Acknowledgements

We thank the reviewers for the thorough comments and suggestions. R.Y. and P.B. gratefully acknowledge support from the National Science Foundation under CAREER award CPS/CNS-1453860 and grants CCF-1837131, MCB-1936775, CNS-1932620 and CMMI-1936624, from the Defense Advanced Research Projects Agency (DARPA) under a Young Faculty Award and a Director Award under grant N66001-17-1-4044, from the Okawa Foundation, from a 2021 USC Stevens Center Technology Advancement Grant (TAG) award, from an Intel Faculty Award and from a Northrop Grumman grant. F.S. is grateful for the support of the NSF under CCF-2106707 and the Wisconsin Alumni Research Foundation (WARF). The views, opinions, and/or findings contained in this article are those of the authors and should not be interpreted as representing the official views or policies, either expressed or implied by the NSF, DoD, or DARPA. The funding sources had no role in study design; in the collection, analysis and interpretation of data; in the writing of this manuscript; and in the decision to submit the article for publication.

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

## A  Additional Theoretical Details

We provide the proofs of the propositions described in Section 4.

First, the proof of Proposition 4.2:

**Proof:**  For our higher-order model, we use a triangle graph $G$ with $G(V) = \{y_1, y_2, y_3\}$. Let the corresponding edge parameters then be $A_{12} = A_{23} = A_{13} := \theta_1$ and $A_{123} := \theta_2$, where $\theta_1, \theta_2$ are non-negative. Suppose that $A_i = 0$ for $i \in \{1, 2, 3\}$. Then,

$$f(y_1, y_2, y_3) = \frac{1}{Z} \exp\left(\theta_1(y_1 y_2 + y_1 y_3 + y_2 y_3) + \theta_2 y_1 y_2 y_3\right).$$

Then, $f(1, 1, 1) \geq f(1, -1, -1)$ and similarly for other combinations with two $-1$'s, and $f(-1, -1, -1) \geq f(1, 1, -1)$ and similarly for other combinations with one $-1$. From this,

$$Z \leq 4 \exp(3\theta_1 + \theta_2) + 4 \exp(3\theta_1 - \theta_2).$$

Using identical reasoning, we obtain a lower bound on $Z$:

$$Z \geq 4 \exp(-\theta_1 + \theta_2) + 4 \exp(-\theta_1 - \theta_2).$$

Then, we have that

$$\begin{aligned}
f(1, 1, 1) &= \frac{1}{Z} \exp\left(3\theta_1 + \theta_2\right) \\
&\geq \frac{\exp\left(3\theta_1 + \theta_2\right)}{4 \exp(3\theta_1 + \theta_2) + 4 \exp(3\theta_1 - \theta_2)} \\
&= \frac{1}{4 + 4 \exp(1 - 2\theta_2)}.
\end{aligned}$$

We also have

$$\begin{aligned}
f(-1, -1, -1) &= \frac{1}{Z} \exp\left(3\theta_1 - \theta_2\right) \\
&\leq \frac{\exp\left(3\theta_1 - \theta_2\right)}{4 \exp(-\theta_1 + \theta_2) + 4 \exp(-\theta_1 - \theta_2)} \\
&= \frac{1}{4 \exp(-4\theta_1 + 2\theta_2) + 4 \exp(-4\theta_1)}.
\end{aligned}$$

Now, consider any lower-order model $f'$. Then,

$$f'(y_1, y_2, y_3) = \frac{1}{Z} \exp\left(\theta_a y_1 y_2 + \theta_b y_2 y_3 + \theta_c y_2 y_3\right).$$

Regardless of how we set the parameters, the symmetry between $(1, 1, 1)$ and $(-1, -1, -1)$ ensures that $f'(1, 1, 1) = f'(-1, -1, -1)$. Then, we have that

$$\begin{aligned}
d_{\mathrm{TV}}(f, f') &\geq |f(1, 1, 1) - f'(1, 1, 1)| + |f'(-1, -1, -1) - f(-1, -1, -1)| \\
&\geq |f(1, 1, 1) - f(-1, -1, -1)| \\
&\geq \frac{1}{4 + 4 \exp(1 - 2\theta_2)} - \frac{1}{4 \exp(-4\theta_1 + 2\theta_2) + 4 \exp(-4\theta_1)}.
\end{aligned}$$

In the second step, we applied the triangle inequality and used the fact that $f'(1, 1, 1) = f'(-1, -1, -1)$.

Now by setting $\theta_1$ sufficiently small and $\theta_2$ sufficiently large, we obtain that

$$d_{\mathrm{TV}}(f, f') \geq \frac{1}{4} - \delta,$$

for any $\delta > 0$. □

To obtain the result on the Dobrushin condition coefficient for a simplicial complex, we can follow the proof of [40], replacing the expressions based on edges with combinatorial sums that involve even or odd numbers of vertices.

## B   Example of Simplicial Complexes

Figure 2 shows an example of a simplicial complex with special properties of no inclusion and no orientation. The incidence matrices are also shown in the figure.

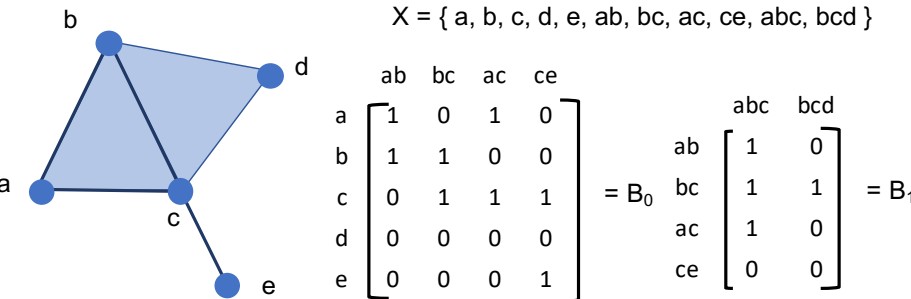

**Figure 2:** An example of unoriented simplicial complex and the incidence matrices. The simplicial complex $X$ consists of 0-simplices $a, b, c, d, e$, 1-simplices $ab, bc, ac, ce$ and 2-simplices $abc, bcd$.

## C   Relationship of Relaxed Simplicial Complexes and Hypergraphs

In this work, we aim at considering a practical and efficient framework to learn irregular higher-order data. We choose to use a relaxed data structure starting from simplicial complexes and we maintain its predefined incidence matrices from $p + 1$ to $p$ dimension as well as the laplacians to determine how relaxed simplices are connected.

The structure of simplicial complexes without orientation and inclusion properties is the same as hypergraphs with additional operations. Traditional simplicial complexes can also be represented as hypergraphs. Sets of hyperedges can be defined by computing the cardinality of each hyperedge to match $p$-simplices. The original incidence matrix of hypergraphs is defined by the set membership of vertices and hyperedges. One can also define and compute the $p$-incidence matrix $B_p$ of hypergraphs for each dimension / cardinality to build the bottom-up and top-down relationship as represented by the incidence matrix of simplicial complexes. But all above requires additional definition and computation from the traditional relationship of hypergraphs.

In contrast, this hierarchical representation is well defined with the formalism of simplicial complexes across different dimensions (between $p$-simplices and $(p + 1)$-simplices). Therefore we choose to define the higher-order adjacency for relaxed simplicial complexes in equation (1) with the help of incidence matrix and laplacian matrix preserved from formal simplicial complexes in this work. The relaxation of properties will not bring in additional computations for the incidence matrix $B_p$. The full adjacency matrix of the connections with arbitrary dimensions described in equation (3) can also be easily written with the help of the incidence matrices $B_p$ and higher-order adjacency matrices $A_p$.

## D   Choices to Build the Full Adjacency Matrix for SCCN

Using the current definition of the higher-order adjacency matrix as in equation (3), one convolutional layer in SCCN will look at the connections of simplices which have the same dimension or the difference is equal to one. For example, the embedding of a $p$-simplex $\sigma$ will be updated with its $p$-simplex, $(p - 1)$-simplex and $(p + 1)$-simplex neighbors (1-hop neighbor) in one convolutional layer. In the next layer, $(p - 2)$-simplex and $(p + 2)$-simplex can also affect its embedding update (2-hops neighbor). If we consider adding the connection of $p$-simplices and $(p + -2)$-simplices in the higher-order adjacency matrix, each simplex can know more in one layer as it has more neighbors.

Considering connections and interactions of simplices with higher dimension difference can help to a more efficient learning if the connections inside $p$-chains are rare and sparse. On the other hand, if the allowed difference is too high, it might lead to a bias learning toward $p$-chains whose cardinality (total number of $p$-simplices) is larger in the dataset.

## E  Latent Representation Learning for Simplicial Complexes

We apply a biased random walk strategy [19] on the binarized version $\bar{A}$ of the higher-order adjacency matrix in Equation (3). Consider a random walk which arrives at simplice $\sigma_t$ and just traverse from $\sigma_{t-1}$, its next step $\sigma_{t+1}$ is generated by the transition probability $P(\sigma_{t+1} = x|\sigma_t, \sigma_{t-1})$. The unnormalized probability $P'$ is given as

$$P'(\sigma_{t+1} = x|\sigma_t, \sigma_{t-1}) = \begin{cases} \frac{1}{p} & \text{if } x = \sigma_{t-1} \\ 1 & \text{if } \bar{A}[x, \sigma_{t-1}] = 1 \text{ and } \bar{A}[x, \sigma_t] = 1 \\ \frac{1}{q} & \text{if } \bar{A}[x, \sigma_{t-1}] = 0 \text{ and } \bar{A}[x, \sigma_t] = 1 \\ 0 & \text{if } \bar{A}[x, \sigma_t] = 0 \end{cases} \tag{8}$$

Let $X$ denotes the simplicial complex (with relaxation of orientation and closure), $walk(\sigma_0)$ denotes the random walk initiated at simplice $\sigma_0$. We apply SkipGram [45] on the random walk of the network data following [19, 44]. The representation mapping $f$ from simplice space to feature space therefore can be learnt by

$$\max_f \sum_{\sigma_0 \in X} \log P\left(walk(\sigma_0)|f(\sigma_0)\right) \tag{9}$$

Compared with existing approaches such as simplex2vec [14] and k-simplex2vec [17], the sc2vec model takes more connections into consideration regardless of the dimension and therefore allows more effective embedding learning.

For simplicity, we set $p = 1$ and $q = 1$ in the experiment section of this work, but the transition probability can also be chosen differently to interpolate between BFS and DFS [19].

## F  Dataset Statistics

**Table 3:** Statistics of simplicial complex datasets. C is the number of classes, D is the dimension of the simplex feature.

| Dataset | C | D | number of $p$-simplices | | | | | |
|---|---|---|---|---|---|---|---|---|
| | | | $p = 0$ | $p = 1$ | $p = 2$ | $p = 3$ | $p = 4$ | $p = 5$ |
| Syn | 2 | 128 | 2,841 | 1,731 | 294 | 25 | 0 | 0 |
| Cora | 7 | 1,433 | 2,481 | 3,590 | 1,294 | 183 | 6 | 0 |
| Pubmed | 3 | 500 | 17,038 | 32,592 | 7,905 | 1,847 | 439 | 173 |
| DBLP | 6 | 1,425 | 899 | 1,672 | 924 | 394 | 0 | 0 |
| DisGene | 26 | 37 | 3,623 | 889 | 422 | 293 | 214 | 156 |
| SBM | 3 | n/a | 600 | 4,771 | 1,969 | 50 | 0 | 0 |
| PPI-BP | 6 | n/a | 1,496 | 3,388 | 3,260 | 1,565 | 302 | 0 |
| HPO-METAB | 6 | n/a | 488 | 3,270 | 5,541 | 5,081 | 2,793 | 861 |

# G  Training settings

We train the models on machine with Intel(R) Xeon(R) CPU E5-2690 v4 @ 2.60GHz. We repeat each experiment of the SCN and SCCN model 100 times with shuffled train/validation/test split. The running time of DHE and K-simplex2vec are up to 20 times as much as the SCN model. Therefore, to make a fair comparison, we allow them 6 more time in duration to finish the experiment. As a result, we run baseline models SIN, MLP, GCN 100 times, DHE 40 times and K-simplex2vec 30 times.

Note that the numbers of simplex are different with different dimension $p$ and the collapsed graph size is also varying from the simplicial complexes. To build a fair comparison, we choose a flexible train ratio $\min\left\{\frac{300}{N_p}, 0.6\right\}$, and validation ratio $\min\left\{\frac{300}{N_p}, 0.2\right\}$ for the SCN framework, where $N_p$ is the number of $p$-simplices. The rest is served as test set. The train/validation/test masks are concatenated for the SCCN model. We also use early stopping to avoid overfitting. Training is skipped if the total number of samples is smaller than 100.

The train ratio for the logistic regression in the comparision of the three latent represenation learning models (sc2vec, simplex2vec, k-simplex2vec) is $\min\left\{\frac{600}{N_p}, 0.6\right\}$, and the rest are used for the test.

**DHE.** The train/validation/test ratio for DHE is the same as the SCCN model in each dimension. The DHE model is complicated and consist of 7 MLP layers. To make a fair comparison, we choose the hidden dimension to be 8 for each fully connected layer in the DHE framework.

**SIN.** The train/validation/test ratio for SIN is the same as the SCCN model in each dimension. SIN is originally designed for graph classification with the inclusion property of simplicial complexes and a maximum dimension of 2. In order to compare with our framework on simplex classification, we consider a variation of the SIN model with two hidden layers. Each layer of each dimension is composed of three MLPs for boundary simplices, upper connected simplices and combination operation. The readout operation is replaced with an output layer to predict the class of each simplex.

**GCN.** We apply a graph convolutional networks [33] with one hidden layer and hidden dimension of 16 on the collapsed graphs from the simplicial complexes and use the node label predictions to predict simplex labels. The collapsed graph is constructed by nodes and edges whose ends are in the same simplex (note that for Cora and Pubmed, it is the same as $A_0$). We train the GCN model with the same train ratio $min\left\{\frac{300}{N}, 0.6\right\}$ and validation ratio, where $N$ is the number of nodes in the collapsed graph. A $p$-simplex $\sigma = [i_0 \ldots i_p]$ is considered as a sample in the baseline test set for $p$-chains if the $(p+1)$ points are all in the graph test set. The simplex prediction is considered as correct if predicted labels of all points in the simplex are true. Note that GCN can only be applied for clique complexes and they cannot be applied to naturally built simplicial complexes (DBLP coauthorship and DisGeNET). Labels of nodes in the collapse graph of a naturally-built higher-order dataset including DBLP co-authorship and DisGene cannot be directly assigned by simplex labels, as the same nodes can exist within several simplices where their labels are different.

**MLP.** we implement a multilayer perceptron model with one hidden layer and hidden dimension of 16. The information of simplex connections are neglected in the MLP model. The train/validation/test masks is the same as the SCCN framework.

**K-simplex2vec and Simplex2vec.** Both of the models automatically include sub-simplices with all the possible combinations of points in simplex. K-simplex2vec can be only applied for $p$-chains when $p > 0$.

# H   Full Accuracy

Accuracy of each simplex dimension is shown in Table 4. We mark the best performances for each dimension $p$ and over all dimensions (last column) separately. While there are several cases when the accuracies for dimension $p$ can be close to each other among multiple models, we mark the highest accuracy as well as the numbers that are close to the best one with at most a difference of $0.005$ in bold.

**Table 4:** Simplex classification accuracy of the methods SCN, SCCN and baselines.

| Dataset | Method | $p=0$ | $p=1$ | $p=2$ | $p=3$ | $p=4$ | $p=5$ | All |
|---|---|---|---|---|---|---|---|---|
| Syn | SCN | 0.6328 | 0.7756 | 0.8175 | n/a | n/a | n/a | 0.6890 |
| | SCCN | **0.6875** | **0.8453** | **0.8677** | **0.9620** | n/a | n/a | **0.7495** |
| | DHE | 0.6257 | 0.7699 | 0.7936 | 0.7950 | n/a | n/a | 0.6822 |
| | SIN | 0.5801 | 0.6241 | 0.7954 | n/a | n/a | n/a | 0.6026 |
| | GCN | 0.6334 | 0.6415 | 0.4788 | 0.2105 | n/a | n/a | 0.6283 |
| | MLP | 0.5448 | 0.6428 | 0.7680 | 0.9500 | n/a | n/a | 0.5858 |
| Cora | SCN | **0.9209** | 0.9603 | 0.9598 | 0.9105 | n/a | n/a | 0.9468 |
| | SCCN | 0.9139 | **0.9743** | **0.9908** | **0.9963** | **1.0000** | n/a | **0.9565** |
| | DHE | 0.7508 | 0.9172 | 0.9391 | 0.9122 | 0.8625 | n/a | 0.8646 |
| | SIN | 0.7181 | 0.8313 | 0.9085 | n/a | n/a | n/a | 0.8089 |
| | GCN | **0.9199** | 0.9516 | 0.8776 | 0.7033 | 0.7143 | n/a | 0.9261 |
| | MLP | 0.7792 | 0.9436 | **0.9861** | **0.9966** | **1.0000** | n/a | 0.8947 |
| Pubmed | SCN | 0.9364 | 0.9541 | 0.9737 | 0.9892 | 0.9926 | 0.9856 | 0.9523 |
| | SCCN | **0.9487** | **0.9727** | **0.9890** | **0.9966** | **1.0000** | **1.0000** | **0.9684** |
| | DHE | 0.8787 | 0.9665 | **0.9909** | **0.9950** | 0.9989 | **1.0000** | 0.9450 |
| | SIN | 0.8898 | 0.9377 | 0.9761 | n/a | n/a | n/a | 0.9304 |
| | GCN | 0.9362 | 0.9522 | 0.9711 | 0.9564 | 0.9579 | 0.9918 | 0.9502 |
| | MLP | 0.8233 | 0.9226 | 0.9792 | **0.9951** | **0.9999** | **1.0000** | 0.9030 |
| DBLP | SCN | 0.6630 | 0.6653 | 0.6858 | 0.6370 | n/a | n/a | 0.6681 |
| | SCCN | **0.7327** | **0.7499** | **0.7882** | **0.8041** | n/a | n/a | **0.7569** |
| | DHE | 0.6643 | 0.6642 | 0.6993 | 0.7111 | n/a | n/a | 0.6738 |
| | SIN | 0.6577 | 0.7095 | 0.7063 | n/a | n/a | n/a | 0.6960 |
| | MLP | 0.7103 | 0.7310 | 0.7787 | 0.7925 | n/a | n/a | 0.7395 |
| DisGene | SCN | **0.4233** | 0.2281 | 0.2353 | **0.2693** | 0.2943 | 0.2903 | **0.3917** |
| | SCCN | 0.3750 | 0.2239 | 0.2004 | 0.2232 | 0.2607 | 0.2619 | 0.3490 |
| | DHE | **0.4234** | 0.1125 | 0.0847 | 0.0903 | 0.0610 | 0.0898 | 0.3679 |
| | SIN | 0.4189 | 0.2315 | 0.2018 | n/a | n/a | n/a | 0.3666 |
| | MLP | 0.3882 | **0.2536** | **0.2460** | 0.2585 | **0.3159** | **0.3387** | 0.3663 |

# I  Accuracy Deviation

We further show the standard deviation of simplex classification accuracy in Table 5. SCN outperforms other methods on DisGene with maximal average accuracy and minimal deviation across the full simplicial complex (column marked with "all"). For synthetic dataset, Cora, Pubmed and DBLP, we observe that SCCN achieves the best performance in accuracy (as shown in the main manuscript), with acceptable low standard deviation of 0.0185, 0.0051, 0.0037, 0.0092, respectively (column marked with "all").

**Table 5:** Standard deviation of simplex classification accuracy of the methods SCN, SCCN and baselines.

| Dataset | Method | $p = 0$ | $p = 1$ | $p = 2$ | $p = 3$ | $p = 4$ | $p = 5$ | All |
|---------|--------|---------|---------|---------|---------|---------|---------|--------|
| Syn | SCN | 0.0164 | 0.0165 | 0.0485 | n/a | n/a | n/a | 0.0117 |
| | SCCN | 0.0187 | 0.0217 | 0.0467 | 0.0834 | n/a | n/a | 0.0185 |
| | DHE | 0.0224 | 0.0416 | 0.0770 | 0.1870 | n/a | n/a | 0.0288 |
| | SIN | 0.0178 | 0.0260 | 0.0641 | n/a | n/a | n/a | 0.0160 |
| | GCN | 0.0165 | 0.0327 | 0.0659 | 0.1584 | n/a | n/a | 0.0219 |
| | MLP | 0.0128 | 0.0217 | 0.0543 | 0.0954 | n/a | n/a | 0.0142 |
| Cora | SCN | 0.0084 | 0.0070 | 0.0108 | 0.0495 | n/a | n/a | 0.0048 |
| | SCCN | 0.0099 | 0.0040 | 0.0040 | 0.0091 | 0.0000 | n/a | 0.0051 |
| | DHE | 0.0197 | 0.0177 | 0.0262 | 0.0591 | 0.2736 | n/a | 0.0161 |
| | SIN | 0.0499 | 0.0347 | 0.0322 | n/a | n/a | n/a | 0.0289 |
| | GCN | 0.0087 | 0.0084 | 0.0232 | 0.0797 | 0.3695 | n/a | 0.0087 |
| | MLP | 0.0087 | 0.0054 | 0.0049 | 0.0089 | 0.0000 | n/a | 0.0054 |
| Pubmed | SCN | 0.0049 | 0.0057 | 0.0048 | 0.0049 | 0.0092 | 0.0202 | 0.0034 |
| | SCCN | 0.0059 | 0.0035 | 0.0024 | 0.0019 | 0.0000 | 0.0000 | 0.0037 |
| | DHE | 0.0187 | 0.0048 | 0.0026 | 0.0045 | 0.0034 | 0.0000 | 0.0077 |
| | SIN | 0.0101 | 0.0073 | 0.0055 | n/a | n/a | n/a | 0.0058 |
| | GCN | 0.0049 | 0.0047 | 0.0061 | 0.0134 | 0.0230 | 0.0111 | 0.0047 |
| | MLP | 0.0039 | 0.0034 | 0.0027 | 0.0017 | 0.0011 | 0.0000 | 0.0030 |
| DBLP | SCN | 0.0227 | 0.0176 | 0.0243 | 0.0615 | n/a | n/a | 0.0131 |
| | SCCN | 0.0191 | 0.0140 | 0.0174 | 0.0447 | n/a | n/a | 0.0092 |
| | DHE | 0.0320 | 0.0283 | 0.0263 | 0.0544 | n/a | n/a | 0.0255 |
| | SIN | 0.0372 | 0.0277 | 0.0272 | n/a | n/a | n/a | 0.0229 |
| | MLP | 0.0201 | 0.0139 | 0.0170 | 0.0458 | n/a | n/a | 0.0106 |
| DisGene | SCN | 0.0058 | 0.0201 | 0.0402 | 0.0578 | 0.0608 | 0.0826 | 0.0058 |
| | SCCN | 0.0475 | 0.0190 | 0.0431 | 0.0555 | 0.0646 | 0.0828 | 0.0382 |
| | DHE | 0.0065 | 0.0199 | 0.0316 | 0.0472 | 0.0452 | 0.0464 | 0.0081 |
| | SIN | 0.0074 | 0.0188 | 0.0414 | n/a | n/a | n/a | 0.0076 |
| | MLP | 0.0214 | 0.0191 | 0.0402 | 0.0509 | 0.0675 | 0.0744 | 0.0174 |

## J    Complexity Analysis

**Table 6:** Complexity analysis: model complexity and duration (wall-clock time, in seconds). $C$ is number of classes and $P$ is the maximum simplex dimension.

| Method | #Hidden dimension | #Hidden Layer | Duration (sec) | | | | |
| --- | --- | --- | --- | --- | --- | --- | --- |
| | | | Syn | Cora | Pubmed | DBLP | DisGene |
| SCN | $16 * (P+1)$ | 1 | 13 | 137 | 447 | 27 | 14 |
| SCCN | 16 | 1 | 10 | 42 | 212 | 13 | 5 |
| DHE | 8 | 7 | 582 | 1,038 | 7,853 | 535 | 784 |
| SIN | $16 * 3 * (P+1)$ | 2 | 26 | 41 | 325 | 36 | 29 |
| GCN | 16 | 1 | 6 | 29 | 45 | n/a | n/a |
| MLP | 16 | 1 | 2 | 34 | 93 | 11 | 5 |

A crucial property for higher-order network models is efficiency. We show the model complexity and average duration (wall-clock time) in Tables 6. The SCCN outperforms other baselines in all datasets except DisGene, with low complexity and reasonable wall-clock time even compared to non-higher order models like GCNs and MLPs.

## K    Model Complexity and Performance

We vary the hidden dimension size in the SCN, SCCN, MLP and GCN, apply the models on Cora dataset and show the trending of prediction performance in Figure 3. Generally, increasing model complexity will deliver better prediction results and achieve a lower deviation. The interquartile range of SCCN with hidden dimension 4 has the similar small size as the hidden dimension increase. The SCN and SCCN have a smaller interquartile range comparing with GCN and MLP, suggesting that they are more robust and stable.

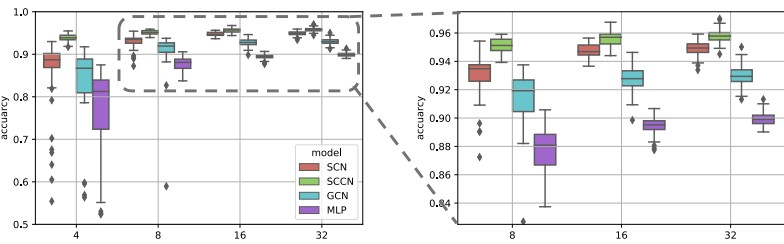

**Figure 3:** We shuffle the dataset and repeat each experiment 50 times with different hidden dimension $4, 8, 16, 32$ on cora dataset and show the boxplot of accuracy.

## L   Sampling DBLP Co-authorship

We observe a explosion of validation loss when applying the SCCN model to the full DBLP dataset [5]. Nevertheless, the distribution of label prediction in the validation set is close to the distribution in the train set and full dataset. This phenomenon suggests that there exists bad samples in the dataset, whose distribution diverges from the major data. To avoid the influence of such data-points, we randomly sample $10\%$ of the DBLP dataset from [5] on each dimension and apply our methods and baselines in the main manuscript. The explosion of validation loss no longer exists with the sampling ratio.

We also vary the sampling ratio from $10\%$ to $100\%$. For each sampling ratio, we randomly sample simplices (papers) existed in the dataset on each dimension $p$ individually, and then we use the subsampled simplices with all dimensions to build the sampled higher-order data. We repeat the experiments 50 times on the DBLP co-authorship dataset. Here we use validation accuracy as the early stopping criteria for the SCN, SCCN and MLP models. The boxplot of accuracy over all dimensions is shown in Figure 4. The performance of the SCN model is enhanced as we sampled more in the co-authorship, while the performance of MLP keeps the similar. We also observe that when the sampling ratio is increasing, the accuracy deviation of the SCCN model is growing, as the number of bad samples is also increasing. The inconsistent trending of performance suggests that more knowledge is encoded in the simplex feature than in the topological structure in this dataset. We also speculate that noisy information is embedded in the connections of simplices with different dimensions, which the SCCN takes into considerations but the SCN does not.

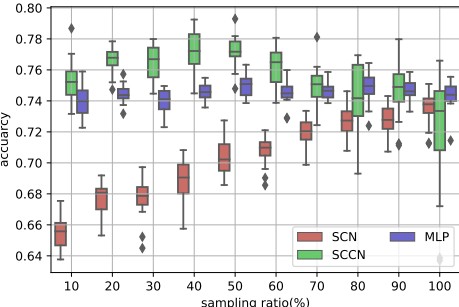

**Figure 4:** We shuffle the dataset and repeat each experiment 50 times with different sampling ratio $0.1, 0.2 \ldots 1.0$ on the DBLP co-authorship dataset and show the boxplot of accuracy.

# M    Citation Prediction

To show the ability of the framework on tasks beyond simplex classification and compare with baselines not applicable to simplex classification, we further apply the sc2vec model to predict the total citation number of co-authorship trained with $30\%$ missing data following the experiment setting in [12, 21, 26]. The latent simplex embedding learnt by sc2vec is served as the input to a neural network with 3 hidden layers with hidden dimension 64. The citation prediction is considered as accurate if the estimation is within $5\%$ of the true value. Beside SNN [12] and SCNN [21], we also consider SAT [27] and SAN [26] as baselines. SAN and SAT both introduce the attention mechanism to simplicial complexes. SAT generalizes the graph attention networks [46] on simplicial complexes with shared attention weights to compute the attention coefficients on upper and down adjacent simplices. In contrast, SAN utilizes two independent masked self-attention mechanisms on lower and upper neighborhoods. The results of the baselines are shown in [26].

The accuracy for each dimension and the number of simplices is shown in Table 7. Our method achieves better performance when $p = 2, 4, 5$. Note that the proposed sc2vec does not directly learn to predict simplex features (citation numbers) as other baselines. The latent embedding of each simplex is first learnt by sc2vec using only the topological structure. The simplex embedding results of sc2vec are later served as the input feature to a neural network which is trained to predict the citation numbers of each simplex. As shown in Table 8, the co-citation dataset has more samples with higher dimensions ($p \geq 2$) compared to the smaller dimensions ($p = 0, 1$). Therefore, sc2vec might be biased towards simplices with higher dimensions as they are more dominant in the dataset, causing better latent embedding learning on simplices when their dimension $p \geq 2$ and relatively worse embedding learning with $p = 0, 1$. In addition, the baseline models directly learn to predict the citation numbers and have different trainable weights for different dimension $p$. As a result, sc2vec performs better when $p \geq 2$ and the result is worse when $p = 0, 1$ compared with the baseline models. In the future, we will consider developing new models which not only provide efficient message flow over the whole simplicial complex without the restriction to the dimension, but can also avoid the potential bias problem.

**Table 7:** Accuracy of citation prediction task. Best ones are marked in bold.

| Dimension | 0 | 1 | 2 | 3 | 4 | 5 |
|---|---|---|---|---|---|---|
| # Simplices | 352 | 1,474 | 3,285 | 5,019 | 5,559 | 4,547 |
| SNN [12] | 0.72 | 0.73 | 0.81 | 0.82 | 0.81 | 0.73 |
| SCNN [21] | 0.72 | 0.73 | 0.81 | 0.82 | 0.81 | 0.74 |
| SAT [27] | 0.19 | 0.33 | 0.25 | 0.33 | 0.47 | 0.53 |
| SAN [26] | **0.75** | **0.89** | 0.82 | **0.94** | 0.95 | 0.96 |
| sc2vec | 0.47 | 0.61 | **0.86** | 0.93 | **0.96** | **0.96** |

