# OpenReview forum: "Efficient Representation Learning for Higher-Order Data with Simplicial Complexes"
_logconference.io/LOG/2022/Conference — LoG 2022 Poster_

### Official Review · Reviewer_aoKd · 2022-10-20

**Overall Score:** 8
**Confidence:** 3

**Review:**

Summary:
The authors introduce novel representation learning models for hypergraphs. The convincingly argue why such higher-order models are necessary and show experimentally that their representation learning models outperform the current state of the art.

Reason for score:
This is a strong paper that introduces novel ideas based on a solid theoretical background. My main concern is that some comparisons with different other methods (both theoretical and experimental) are missing and that the presentation is at times confusing.

Strong points:
- The work is based on solid theoretical insights.
- The representation learning models SCN, SCCN and sc2vec are novel.
- Authors performed a diverse set of experiments with reasonable baseline models.

Weak points:
- A comparison with Simplicial Graph Attention Network (reference 12) and Simplicial Neural Networks (reference 13), both in methodological differences and in the experiments, would have strenghtened the results.

- The presentation is unclear. In particular:
  + It is very confusing that the authors chose to call the hypergraphs for simplicial complexes, even tough they are not closed under taking subsets.
  + The writing is at times very non-linear. For example lines 150-157, to understand the formula (2) in line 150, the reader has to read a whole paragraph before all the symbols are explained.
  + Inconsistent notation. For example Figure 2, in the text (line 209) uses notation {1, 2, 3}, while in appendix B it uses notation {a, b, c}.
  + Some terms are not explained. For example what is d_TV on line 223?
  + What does the bold face mean in the tables?
  + The order of the models is different in tables 1&6 versus tables 4&5.
  + In the reference to Figure 2 (line 210), it is not mentioned that this is in the appendix.
  + Typo: A word is missing in line 508.

---

### Official Review · Reviewer_XwnR · 2022-10-20

**Overall Score:** 3
**Confidence:** 4

**Review:**

Summary:

The paper proposes an abstraction to simplicial complexes, renouncing closure under taking subsets, and simplex orientation, in lieu of scalability and ease of interpretation. Building upon the novel encoding of higher dimensional interactions, three models of inference are proposed, Simplex Convolutional Networks (SCN), that generalize graph convolution to p-chains, Simplicial Complex Convolutional Networks (SCCN), that generalize graph convolution to the whole simplicial complex construction, and finally, sc2vec, a simplex embedding method akin to node2vec, but build on a novel adjacency matrix that stems from the proposed simplicial complex construction. Due to the construction of the adjacency matrix, the latter two models consider all simplices of the defined complex, regardless of dimension.  Furthermore, a higher-order data model is proposed that incorporates the higher-dimensional interactions naturally occurring in datasets. The model is subsequently used to derive enhanced expressiveness and generalization abilities gained by considering higher-order interactions in the form of simplicial-complex-encoded hypergraphs. Specifically, a total variation bound, as well as a bound on influence for satisfying Debrushin's condition of weak independence, are derived.


Review:

The strong points of the paper are as follows:
- A compelling argument is presented against encoding all subsets of higher-order interactions, as necessitated by the standard simplicial complex construction. Instead, a novel construction is presented that relaxes key aspects of simplicial complexes, namely, subset under closure and orientation information.
- Built upon the "generalized" simplicial complex construction, a full learning suite is proposed, considering convolutional networks of varied granularity, and simplex embedding strategies.
- The theoretical contributions regarding the expressiveness and generalization abilities of the higher-order data model presented are interesting and novel, to the best of my knowledge, as they diverge from the usual Weisfeiler-Lehman test comparisons with standard graphs, and adopt a more functional analysis perspective.
- Experimental evaluation of the proposed models against a variety of related baselines and different datasets, with competitive performance.

Nevertheless, some major concerns arise regarding the clarity of exposition and the proposed relaxation to simplicial complexes. Namely:
1. The paper feels like it is lacking focus, with a plethora of results and models presented that consider the proposed simplicial complex construction, hypergraphs, and higher-order interactions, with no clear distinction.
2. According to the proposed simplicial complex construction subset inclusions can be discarded, which may lead to some problematic cases. Let a collection of edges in the likes of the example appearing on the top left of Figure 1 $X_0=\\{ab, bc, cd, de, ae\\}$, encoding a cycle. In this case all adjacency and incidence matrices $A_0, A_1$ and $B_1$ should be zero, since the faces of the edges are not included in the complex. This has the effect of a trivial convolution and trivial embeddings for all models, since no information is diffused among the simplices.
3. Altering incidence relations by voiding subset inclusion invalidates the boundary condition $B_{k-1}B_k=0$, implying in turn that homological information encoded in standard simplicial complexes, and their Hodge Laplacians, is lost.
4. In line 143 it is stated that when $p=0$, $A_0$ would be equivalent to the adjacency matrix of a regular graph, yet that would not be the case for the top left example of Figure 1 ($A_0=0$).
5. The details of the proposed sc2vec latent representation strategy are not presented.
6. The graph-structured data model proposed in Section 5.1, as well as the subsequent theoretical analysis, eventually stems from hypergraph representations, as noted at the end of the same section.
7. Since one of the main contributions of the proposed model is scalability, it would be interesting to compare the memory footprint of the models and data structures against other higher-order alternatives.
8. It would be interesting to examine and compare the influence of the number of layers on the higher-order convolutional models considered.

Questions:
1. Considering the above comments, what are the benefits of using the proposed relaxation of simplicial complexes, instead of hypergraphs (in which context the theoretical analysis is developed, and proposed models would still be valid)?
2. What would be the significance and interpretation of the spectrum of the Laplacian that arises from the proposed generalized simplicial complex construction that ignores closure under subsets and orientations?
3. How is the diagonal of matrix $A_p$ defined in equation (1)?
4. How is the good performance of MLP explained, especially for high p values (Figures 4 & 5), in the context of the theoretical insights of Section 5, which imply that appropriate models encoding higher-order information are necessary to ensure expressiveness and generalization?
5. In Appendix I a validation loss explosion is observed, which is attributed to a noisy dataset. This loss explosion also manifests as the sampling ratio is increased in Figure 4. How is the nearly constant performance of MLP explained in this experiment? Shouldn't the peculiarities of the dataset affect MLP's performance as well?

Typos:
- In section 4.2 variable $D_p$ is used to indicate both the embedding dimension and the normalization matrix.
- In equation (4) $\theta$ should not be subscripted in order to match Daskalakis' formulation [29]. Furthermore, in the same equation, doesn't $A$ encode adjacency information?

While some theoretical contributions are significant and novel, and the experimental evaluation overall indicates competitive performance, the aforementioned downsides drive me to suggest rejection of the paper in its current form. Hopefully the Authors' response will clear up any confusion regarding the proposed simplicial complex construction.

---

### Official Review · Reviewer_pvpq · 2022-10-21

**Overall Score:** 5
**Confidence:** 4

**Review:**

The manuscript "Efficient Representation Learning for Higher-Order Data with Simplicial Complexes" is a submission to the conference's 9-page track. The authors adhere to the page limits and to the LaTex template provided by the conference.

**Goal of the paper and summary of contributions**

The goal of the paper is to generalize graph convolutional networks to relational data capturing non-dyadic interactions as in, e.g., co-authorship or co-occurrence data. Capturing non-dyadic interactions based on higher-order matrix representations of graphs, the authors develop two higher-order generalizations of graph convolutional networks, simplex convolutional networks (SCNs) and simplicial complex convolutional networks (SCCNs). The key idea behind those models is to consider p-simplices separately in the message passing, which is achieved by defining a higher-order "supra"-adjacency matrix that contains, as elements, adjacency matrices for p-simplices separately. They further propose a generalization of node2vec to this higher-order matrix, which they call sc2vec. The authors evaluate the performance of those models in four empirical and one synthetic data set on graphs with non-dyadic relationships and address the higher-order graph learning task of simplex classification.

**Strong points**

- Investigating a generalization of deep graph learning to data with higher-order relationships, the paper addresses an important topic. The method shows improved performance over existing higher-order graph neural networks and could thus have impact on practical applications of graph learning in data capturing non-dyadic interactions.

- The paper is generally well-written and it is (mostly) easy to follow the authors' arguments. The paper opens with a clear and concise motivation and it includes didactic examples that illustrate key ideas behind the method.

- Going beyond standard node- or edge-centric learning tasks, the authors consider the higher-order graph learning task of *simplex classification* and show that their method outperforms the best competing baseline. The authors further tested their method in a citation number prediction task.

- The paper includes both theoretical and experimental results. The experimental setting is described in detail and the experiments seem to have been executed in a solid and reproducible way. The appendix includes data set statistics, details on the experimental setup and training settings, detailed results for different p-simplices, standard deviations of results, as well as wall clock times.

**Weak points**

- The contribution of the paper over existing methods that generalized message passing to higher-order interactions should be clarified, especially the research gap motivating the present work and the contribution of the model with a relaxed closure assumption over existing hypergraph neural networks (see questions 1 and 2).

- While the paper is generally well-written, there is potential to improve the description of some aspects of the methodology (see questions 3, 4, 9, 10).

- It is not clear how the simplifying assumption of exclusively considering relationships between p-simplices whose dimensions differ by at most one affects the results (see question 5).

- While the inclusion of a simplex classification task is a strength of the work, it would still be interesting to compare the performance of the proposed models in node-centric classification or community detection tasks (see question 6).

- While the appendix features results for a citation count prediction task, at the node and edge level the performance of s2vec is much worse than that of the competing methods (see question 7)

- For the theoretical results in section 5, a hypergraph model is used to "simulate" simplical complexes. While this can be used to generally reason about the need for higher-order graph models, I do not see which insights it brings for the specific models proposed in the paper (see question 8).

- The authors do not mention whether their code will be made available.

**Detailed suggestions and questions for authors**

1) In section 2, the authors write that existing models were ...

"mainly examined on data which is built by uplifting graphs or images to clique complexes, and the learning tasks still focus on graph-based problems. Clique complexes are only analog of real-world complex higher-order information and it is hard to explain their physical meaning. The properties of simplicial complexes such as orientation and inclusion also make the approaches difficult to apply to real-world datasets with natural higher-order structures beyond clique complexes lifted from graphs."

I think it would be helpful to further clarify the research gap that motivates the authors' approach, specifically the following points:

- What do you mean by "data which is built by uplifting graphs or images to clique complexes")? Prior (cited) works on simplicial complexes have partly used the same data as those in the current manuscript (e.g. DBLP coauthorship data) which directly provide information on non-dyadic relationships.

- Why do "orientation and inclusion [...] make the approaches difficult to apply to real-world datasets". Could you provide more details what exactly makes this difficult? Since inclusion (which I assume refers to downward closure?) is a feature of simplicial complexes, couldn't this be addressed by using a hypergraph-based model (which do not assume downward-closure)? Part of this is seemingly explained in section 3.2 but I could not follow the argument there (especially how this would invalidate hypergraph models).

I generally would like to learn more about the relation between the proposed relaxed simplicial complex model and prior works on hypergraph neural networks, which are used as baseline in the evaluation but not mentioned in the related work section.

2) I was wondering how the relaxed definition of simplicial complexes without subset/downward closure is different from hypergraphs? Wouldn't the representation of coauthorships as multiple hyperedges with different cardinalities also address the issue that the authors outline in the example in section 3.2?

3) Please clarify the definition of a p-chain in section 3.1. One can define p-chains as formal sums of p-simplices with different coefficients, so which assumptions on those coefficients do you make? Do you consider p-chains as simple collections of some of the p-simplices (i.e. using binary coefficients 0 or 1 for the simplices)? Please clarify.

4) Could the authors clarify the process of building "clique complexes", which is mentioned in section 2 and 6.1? Does this refer to replacing all cliques of size k in a graph by a corresponding k-1 simplex?

5) In section 4.2, the authors highlight that the proposed definition of the higher-order adjacency matrix ignores relationships between simplices whose dimensions differ by more than one. It is further stated that this could be easily addressed with an updated definition. Could you clarify how this simplifying assumption could influence the results? What do we lose due to this simplification compared to previous works that have not consider it? What do we gain?

6) While I appreciate the evaluation of the method in a higher-order graph learning task (i.e. simplex classification) it would still be interesting to know if and how the proposed method works for node- or graph-centric learning tasks like node or graph classification or community detection.

7) It is curious that the performance of s2vec in the citation count prediction at the node and edge level (p=0 and p=1) (appendix J) is much worse than for the competing methods. Can this be explained?

8) I think there is potential to better integrate and explain the theoretical results in section 5 in the context of the specific models proposed in the paper. What insights can we get from the hypergraph-based model in (4) for the simplicial complex models proposed in the paper? How is this related to the author's relaxation of the downward closure? And what do we learn from the results in 5.3, arguing that we cannot use non-graphical models to model graph-structured data? How does this finding relate to the state-of-the-art in statistical learning theory? Is it central to the paper or would it more reasonable to move it to the appendix?

9) In Table 1, there are no results for GCN in the DBLP and DisGene data. In the appendix, the authors write:

"Note that GCN can only be applied for clique complexes and they cannot be applied to naturally built simplicial complexes (DBLP coauthorship and DisGeNET) because they add all possible sub-simplices in the simplicial complex"

However, in section 6.2  it is stated that the GCN model is just applied to the collapsed graph, which should be possible for the DBLP network data. Please clarify.

10) I would suggest to include a more detailed explanation of sc2vec. How do you simulate random walks on the simplicial complex?

**Typographical suggestions**

- line 62: "within _the_ last year"
- line 63: _the_ attention mechanism ... and _the_ Hodge Laplacian
- line 303: where inside points have _the_ same label
- line 588: train_ed_ with 30 % missing data

---

### Official Review · Reviewer_km56 · 2022-10-21

**Overall Score:** 1
**Confidence:** 5

**Review:**


Many parts of this work ideas/equations/normalization/proposed method is essentially proposed elsewhere and discussed by various authors [1,2,3,4,5]. Specifically the conv operation that proposed is exactly suggested in with the exact same normalization that the authors are claiming to have [1,3,5]. In particular, the same normalization/conv network claimed by the author is proposed in [5]. More importantly simplicial complex representation learning is studied in [3,5]. The authors seem to have neglected to do a proper prior search on the related work and the above citation are never mentioned.  More importantly, there is an entire paper called "simplicial complex representation learning" [1]. Many of the proposed ideas are discussed there in details.


Other ideas in the paper discussed elsewhere : working with incomplete simplicial complex is discussed in [2]

The only section that seems to be original and not discussed previously is 5.3. Otherwise, I found the paper to massively ignores relevant and sometimes identical literature. Since many of the networks suggested here are identical to the earlier proposed methods without proper citation, I reject the paper.


other issues : for the sake of reproducibility, the authors should provide more details about the training configurations and hyperparameters selection.

Here are my suggestions to the paper for future submissions :

1- properly do a lit review and properly and precisely give credit to prior work, mentioned here and otherwise. Simple google usually does a lot. I Googled simplicial complex representation learning and I found many of the refences I provided here. The author should not have missed that.  If your work is very close to existing work, have a section dedicated to that prior work, show its weaknesses and explain how your solution is better, explain why and explain the differences and the similarities, demonstrate with experiments your solution is superior.

2- the conv net proposed here is identical  to the one originally proposed [5,3], explain this. Expand your method and make it at least slightly different and explain the why your slight difference might give marginal improvement by giving a specific example where that happens. While this is incremental, it at least abides with the typical academic standards.

3- Compare your rep learning on simplicial complex method against existing very very similar uncited work [1,5], try to distinguish your work by making your work different in many aspects so that it diverges from existing work and it has its unique features. Again explain similarity and differences, demonstrate your method with more experiments.

4- Explain that prior work exist when it comes to considering incomplete complexes [2], show what you did different, what did you improve upon existing work?

5- Emphasis more the only novel part of the work which is 5.3 and expand more on this section--it has interesting results that I did not see elsewhere.

While suggestions 1,2,3,4 are going to make the work incrementally better than existing work, (and I think more emphasis should be put towards 5) it will integrate better in the larger existing literature on this topic.

Given the above, while I think there is a a very small novel component to the paper (section 5.3), the vast majority of the proposed work has been discussed elsewhere without proper citation/discussion/analysis/comparison/, I recommend rejection of this paper.
________________

missing refs

[1] @article{hajij2021simplicial,
  title={Simplicial Complex Representation Learning},
  author={Hajij, Mustafa and Zamzmi, Ghada and Papamarkou, Theodore and Maroulas, Vasileios and Cai, Xuanting},
  journal={Machine Learning on Graphs (MLoG) Workshop at 15th ACM International WSD Conference},
  year={2022}
}

[2] @article{hajij2022higher,
  title={Higher-Order Attention Networks},
  author={Hajij, Mustafa and Zamzmi, Ghada and Papamarkou, Theodore and Miolane, Nina and Guzm{\'a}n-S{\'a}enz, Aldo and Ramamurthy, Karthikeyan Natesan},
  journal={arXiv preprint arXiv:2206.00606},
  year={2022}
}

[3] @article{bunch2020simplicial,
  title={Simplicial 2-complex convolutional neural nets},
  author={Bunch, Eric and You, Qian and Fung, Glenn and Singh, Vikas},
  journal={arXiv preprint arXiv:2012.06010},
  year={2020}
}




[4] @inproceedings{hajij2022high,
  title={HIGH SKIP NETWORKS: A HIGHER ORDER GENERALIZATION OF SKIP CONNECTIONS},
  author={Hajij, Mustafa and Ramamurthy, Karthikeyan Natesan and Saenz, Aldo and Zamzmi, Ghada},
  booktitle={ICLR 2022 Workshop on Geometrical and Topological Representation Learning},
  year={2022}
}

[5] @article{hajijcell,
  title={Cell Complex Neural Networks},
  author={Hajij, Mustafa and Istvan, Kyle and Zamzmi, Ghada},
  journal={NeurIPS 2020 Workshop TDA and Beyond},
  year={2020}
}

---

### Meta-Review · Area_Chair_pfyz · 2022-11-17

**Confidence:** 4
**Recommendation:** Accept

**Meta Review:**

Summary of review comments:

1 strong reject - Reviewer km56: gave negative recommendation based on novelty claims, and lack of an accurate presentation of related work. Based on interaction between authors and reviewers, I believe the paper now appropriately addresses related work. However, I advise the authors to include in a final revision explicit mention of the similarity to CCXN, other than that I think the account of related work is just. My sincere thanks to Reviewer km56 for paying attention to this.

5 weak reject - Reviewer pvpq: raised related remarks regarding clarification with respect to novelty over related work. After several revisions this has done this appropriately.

3 clear reject - Reviewer XwnR: Also gave constructive feedback and formulated initial concerns which have been addressed by the authors by clarifying the paper and formulating the right scope of it.

9 clear accept - Reviewer aoKd: Also gave constructive and critical feedback which was taken into account by the authors.

In summary I believe many of the criticism challenged claims made by the authors, where doubts by reviewers often found their origin in ineffective exposition of the method and setting a clear scope, also in relation to other works. The authors’ response address these and give credibility to the paper; they significantly improved the paper. After these revisions and thorough checks by the reviewers (thank you!) I consider the paper to be sound.

Also, based on the feedback I think related work is appropriately discussed.

I think the paper in its current form -based on the excellent suggestions by all reviewers and the effort by the authors- is of good quality. As such I recommend accept.

---

### Decision · Program_Chairs · 2022-11-22

**Decision:**

Accept (Poster)

**Comment:**

We discussed this paper among the PCs and agree with the assessment of the AC. We strongly encourage the authors to take into account the suggestions for framing the claims wrt. related work but find that all major technical issues have been rectified during the rebuttal phase.